# Pulse-Echo Ultrasonic Verification of Silicate Surface Treatments Using an External-Excitation/Single-Receiver Configuration: ROC-Based Differentiation of Concrete Specimens

**DOI:** 10.3390/ma18163765

**Published:** 2025-08-11

**Authors:** Libor Topolář, Lukáš Kalina, David Markusík, Vladislav Cába, Martin Sedlačík, Felix Černý, Szymon Skibicki, Vlastimil Bílek

**Affiliations:** 1Faculty of Civil Engineering, Brno University of Technology, Veveří 331/95, 602 00 Brno, Czech Republic; 2Faculty of Chemistry, Brno University of Technology, Purkyňova 464/118, 612 00 Brno, Czech Republic; kalina@fch.vut.cz (L.K.); david.markusik@vut.cz (D.M.); vladislav.caba@vut.cz (V.C.); bilek@fch.vut.cz (V.B.); 3Faculty of Civil and Environmental Engineering, West Pomeranian University of Technology in Szczecin, al. Piastów 17, 70-310 Szczecin, Poland; szymon.skibicki@zut.edu.pl

**Keywords:** pulse-echo ultrasonic testing, non-destructive evaluation, surface treatment identification, silicate sealer, ROC analysis, cement-based materials, quality control

## Abstract

This study investigates a non-destructive, compact pulse-echo ultrasonic method that combines an external transmitter with a single receiving sensor to identify different surface treatments applied to cementitious materials. The primary objective was to evaluate whether treatment-induced acoustic changes could be reliably quantified using time-domain signal parameters. Three types of surface conditions were examined: untreated reference specimens (R), specimens treated with a standard lithium silicate solution (A), and those treated with an enriched formulation containing hexylene glycol (B) intended to enhance pore sealing via gelation. A broadband piezoelectric receiver collected the backscattered echoes, from which the maximum amplitude, root mean square (RMS) voltage, signal energy, and effective duration were extracted. Receiver operating characteristic (ROC) analysis was conducted to quantify the discriminative power of each parameter. The results showed excellent classification performance between groups involving the B-treatment (AUC ≥ 0.96), whereas the R vs. A comparison yielded moderate separation (AUC ≈ 0.61). Optimal cut-off values were established using the Youden index, with sensitivity and specificity exceeding 96% in the best-performing scenarios. The results demonstrate that a single-receiver, one-sided pulse-echo arrangement coupled with straightforward amplitude metrics provides a rapid, cost-effective, and field-adaptable tool for the quality control of silicate-surface treatments. By translating laboratory ultrasonics into a practical on-site protocol, this study helps close the gap between the experimental characterisation and real-world implementation of surface-treatment verification.

## 1. Introduction

Pulse-echo ultrasonic testing (PEUT) is a widely used non-destructive evaluation (NDE) technique for assessing the internal condition of concrete structures. In PEUT, a single ultrasonic transducer emits high-frequency acoustic pulses into a material and receives echoes reflected from internal interfaces, voids, defects, or anomalies. Its major advantage is its applicability to members that are accessible from only one side because emission and reception occur on the same surface [1,2,3,4].

The method operates by measuring the round-trip travel time of the pulse until it is reflected from an internal feature and returned to the transducer. With a known wave velocity, this permits depth-resolved localisation of anomalies and thickness gauging without damage [2,3,4,5,6]. Recent field reports confirm that travel-time-based pulse-echo imaging can reveal delaminations, honeycombing, and reinforcement cover variations in bridge decks and tunnel linings with sub-centimetre accuracy [7].

Conventional PEUT instruments either toggle one element between the transmit and receive modes or use a closely spaced dual-element probe. However, the ring-down of the transmitter often masks the early time surface reflections and increases the dead zone length. Therefore, several groups have separated the emission and reception functions while retaining one-sided access by placing a low-cost external actuator next to (but electrically isolated from) a broadband receiver. Laboratory studies have shown that such external-excitation/single-receiver (Ext-Tx/1Rx) pulse-echo rigs can match the flaw-detection capability of phased arrays once modern de-ringing and peak-picking algorithms are applied [7,8].

In addition to detecting bulk flaws, PEUT is intrinsically sensitive to acoustic-impedance discontinuities within the first few millimetres of the surface, where each interface acts as an additional reflector and perturbs the earliest back-wall echo [7]. Numerical and experimental studies on concrete skins, battery casings, and ice layers have shown that when the local impedance contrast increases above ≈ 10%, a thin coating (or any densified zone) can raise or lower the first-echo amplitude—and shift its dominant frequency—by ≥ 6 dB. Such sensitivity underpins the present work: silicate sealers precipitate C-S-H gels that stiffen the near-surface matrix and therefore increase impedance; however, robust ultrasonic acceptance criteria for these treatments are still lacking [8,9].

Amplitude-based descriptors, such as the maximum amplitude, RMS voltage, energy, and rise time, have proven reliable for differentiating adhesive debonding in non-destructive analysis, near-surface voids in reinforced decks [7], and roughness-controlled interfaces [10,11]. Autoencoder post-processing and analytic-signal suppression of front- and back-wall echoes further enhance the discrimination in the first ply or cover zone.

### Unresolved Issues in Current Research

One-sided quality assurance of surface treatments: Existing synthetic aperture focusing technique (SAFT)-based scanners still entail bulky arrays and extensive scanning, hampering on-site quality control on vertical or overhead faces [12,13].Objective decision thresholds: Many case studies remain qualitative, and ROC-based pass/fail criteria are largely absent for protective coatings [14,15,16].Integration of modern classifiers: Only a few studies have combined single-sensor PEUT with machine-learning classifiers for concrete applications [8].

Therefore, this experimental program investigated whether a simple pulse-echo setup using an external source and one receiving sensor, analysed through four straightforward amplitude metrics—peak amplitude, RMS voltage, signal energy, and rise time ratio—could reliably distinguish untreated mortar prisms (R) from specimens created with two commercial lithium water-glass sealers (A and B). ROC analysis was employed to benchmark the discrimination performance and derive practical acceptance thresholds. By focusing on surface-induced impedance contrasts rather than deep-seated flaws, the study offers practitioners a fast, inexpensive, and fully non-destructive tool for the on-site verification of silicate treatments, thereby addressing a gap highlighted by recent reviews.

## 2. Materials and Experimental Setup

Cement mortar specimens in the shape of prisms were prepared following a modified version of the standard procedure described in [17]. The ordinary Portland cement used was CEM I 42.5 R, supplied by the HeidelbergCement Group (Mokra plant., Mokrá-Horákov, Czech Republic). The water-to-cement ratio (*w*/*c*) was 0.80, and the aggregate-to-cement mass ratio was 4:1 using standard siliceous CEN sand. These mixing ratios intentionally deviated from the standard recommendations. Higher ratios were selected to create a more porous cementitious system [18,19], thereby enhancing the observable effects of the surface treatment.

After casting, the specimens were demoulded after 24 h and subsequently cured in water for 28 d to ensure a sufficient degree of hydration. Following this phase, the samples were removed from the water bath and conditioned for an additional 48 h under laboratory conditions (approximately 25 °C, RH ~40%).

The surfaces of the mortar specimens were then treated with the selected silicate sealers (Table 1). Prior to application, the surface was mechanically prepared by dry grinding with a diamond grinding disc to remove the carbonated surface layer and ensure better penetration of the treatment agents. Once the surface was thoroughly cleaned, the specimens were immersed in a bath containing the respective sealer for 24 h, ensuring uniform exposure on all sides. The reference specimen (R) was immersed in deionised water under the same conditions. After the immersion period, any excess liquid was removed from the surface using a brush, and the specimens were stored under laboratory conditions (25 °C, RH ~40%) for 1 week before non-destructive testing.

Non-destructive experimental measurements were conducted using a through-transmission ultrasonic configuration to evaluate the effects of surface treatment on cementitious materials. The specimens were prismatic cement-based composite specimens with dimensions of 40 mm × 40 mm × 160 mm. Three prisms were tested for each series; on each prism, five tone bursts were recorded on two mutually orthogonal faces, yielding three prisms × five pulses × two faces = thirty independent scans for the statistical evaluation. Each specimen was positioned on a profiled polyurethane foam pad to minimise the reflections and acoustic coupling with the support surface.

A custom broadband actuator (Dakel, Ø 12 mm active face, −3 dB = 250 kHz–7 MHz, TVR = 0.55 mV Pa^−1^ at 5 MHz) was clamped to one face of each prism and driven by a generator Keysight 33220A (Agilent, Santa Clara, CA, USA, 2023) to emit a five-cycle 5 MHz tone-burst (Table 2). A larger aperture ensured a nearly planar wavefront across the entire specimen face. The transmitted field was captured on the opposite face using a commercial IDK-09 receiver (ZD Rpety-Dakel, Ø 9 mm body, Ø 6 mm PZT-200 ceramic). Although its nominal flat response spans 100–500 kHz, reliable operation up to 1 MHz has been reported; therefore, we accepted a ≈ 12 dB sensitivity loss at 5 MHz in exchange for a higher axial resolution. Both probes were held by clamps and coupled with a ~0.1 mm film of Vaseline petroleum jelly (Figure 1).

The received waveform was digitised by a 16-bit A/D converter DAKEL ZEDO-BOX (ZD Rpety-Dakel from Prague, Czech Republic) at 10 MHz, yielding 125,300 samples per record (12.53 ms trace length) with an analogue front-end gain of +34 dB; raw signals were stored as little-endian, signed 2-byte integers. Representative exported data are openly available on Zenodo at https://doi.org/10.5281/zenodo.15860905 (accessed on 1 August 2025).

All measurements were performed under laboratory conditions at 21 ± 1 °C and a relative humidity of 45 ± 5%. The photo illustrated in Figure 2, showing the prismatic mortar specimen clamped between the transmitting and receiving transducers on a profiled polyurethane foam pad, provided stable coupling and highly repeatable through-transmission measurements, enabling the rapid assessment of surface treatments in cementitious materials.

## 3. Evaluated Parameters and Statistical Approach

In this study, a comprehensive set of ultrasonic signal parameters was analysed to assess the effectiveness of surface treatments on cementitious materials using the pulse-echo method. The selected parameters focused on time-domain features, which provided insights into the interaction between ultrasonic waves and the treated and untreated surfaces.

Time-Domain Parameters

Maximum amplitude (V): Represents the peak voltage of the received ultrasonic signal, indicative of the energy reflected from internal features or surface modifications.Root mean square (RMS) (V): This measures the signal power over time and reflects the overall energy content of the reflected wave.Signal duration (ns): This denotes the time interval during which the signal maintains a significant amplitude, indicating the damping characteristics and potential scattering in the material.Rise time (ns): The time required for the signal to rise from a defined low percentage to a high percentage of its peak amplitude, which is used to assess the sharpness of the signal.Energy (V2/Hz): Obtained by integrating the squared voltage spectrum over the analysis bandwidth and expressing the result per unit frequency, thus providing a bandwidth-normalised measure of the ultrasonic pulse energy.

Ratio-Based Parameter

Rise time to duration ratio (–): Provides a normalised measure of the signal onset sharpness relative to its total duration, potentially highlighting differences caused by surface treatment.

The selection of these parameters was based on established ultrasonic testing practices, wherein variations in the time-domain features reflect changes in the density, elasticity, and surface conditions [20,21,22].

To determine the statistical significance among the specimen groups, the following analytical methods were used:Descriptive statistics: Computation of the mean, median, standard deviation, and interquartile range for each parameter.Normality tests: The Shapiro–Wilk test was applied to assess the data distribution.Inferential statistics: Depending on the data distribution, either the Student’s *t*-test, Welch’s *t*-test (Welch), or the non-parametric Mann–Whitney U test (MW) was used for between-group comparisons.Classification modelling: Logistic regression models were built to predict the probability of a specimen being treated or untreated, based on the evaluated parameters. Model performance was assessed using receiver operating characteristic (ROC) curves and the area under the curve (AUC) metric. The optimal cut-off values were derived using the Youden index.

This statistical approach enabled a robust evaluation of the ultrasonic parameters and their effectiveness in distinguishing the surface-treated from untreated specimens, thereby supporting the development of a practical, non-destructive diagnostic method [23,24,25].

## 4. Results

### 4.1. Max-Amplitude (V)

The maximum amplitude of the reflected ultrasonic signal differed significantly among the three groups: reference R and treatments A and B. As illustrated in Figure 3, Group A showed the highest amplitudes (mean = 316.2 µV), followed by the Reference Group R (mean = 234.7 µV), while Group B exhibited the lowest values (mean = 166.4 µV), indicating a marked signal attenuation.

The summary statistics for each group, including the mean, standard deviation, skewness, and 95% confidence intervals, are shown in Table 3.

Statistical testing confirmed that these differences were highly significant for all pairwise comparisons. Both the Mann–Whitney U test and Welch’s *t*-test yielded *p*-values well below the 0.001 threshold:R vs. A: *p* = 1.56 × 10^−5^ (MW), *p* = 3.29 × 10^−6^ (Welch);R vs. B: *p* = 2.66 × 10^−8^ (MW), *p* = 2.40 × 10^−7^ (Welch);A vs. B: *p* = 1.65 × 10^−9^ (MW), *p* = 1.49 × 10^−12^ (Welch).

The boxplot in Figure 3 clearly reflects the statistical findings, showing well-separated amplitude distributions for each treatment group. Notably, Group B consistently displayed the lowest signal amplitudes, suggesting increased acoustic attenuation or impedance mismatch at the surface interface. Conversely, the specimens in Group A demonstrated amplitudes that exceeded those of the untreated reference, which may be attributed to improved acoustic coupling or a smoother surface profile resulting from the treatment.

These results underscore that the maximum amplitude is a highly sensitive and reliable parameter for distinguishing surface treatments in cementitious materials using the pulse-echo ultrasonic method.

#### Amplitude ROC Analysis

To assess the diagnostic capability of the maximum amplitude as a classification parameter, a receiver operating characteristic (ROC) analysis was performed for all three pairwise group comparisons (see Table 4): the untreated reference (R), silicate-treated specimens (A), and sealer-treated specimens (B). 

R vs. A (Figure 4)

The ROC curve yielded an AUC of 0.83, indicating the good discriminative power of the amplitude feature in separating silicate-treated samples from the reference group. Applying a cut-off value of 2.89 × 10^−4^ V resulted in both sensitivity and specificity values of 83%. This suggests that treatment A consistently increased the reflected amplitude relative to the untreated surfaces, supporting the applicability of the amplitude as a screening parameter for surface enhancement detection.

B vs. R (Figure 5)

In this comparison, class inversion was required because of the reversed directionality of the amplitude (lower in Group B). After recoding, the analysis yielded a corrected AUC of 0.93, demonstrating excellent separation. The optimal cut-off of 1.98 × 10^−4^ V provided a sensitivity of 90% and specificity of 89%. This confirms that treatment B significantly reduced the acoustic signal amplitude compared with the reference, likely due to the increased damping or altered surface impedance.

B vs. A (Figure 6)

The most distinct separation was observed between the two treatment groups with an AUC of 0.97. The optimal decision threshold of 2.04 × 10^−4^ V achieved a perfect sensitivity (100%) and specificity of 93%, indicating that the amplitude is a highly reliable parameter for distinguishing between the two surface modification protocols. Silicate treatment A yielded consistently stronger reflections, whereas the B-type treatment introduced variability and signal attenuation.

These findings demonstrate that the maximum amplitude is not only effective for identifying surface treatment in general, but is especially well-suited for the binary classification of distinct treatment modalities. Its robustness across all comparisons, particularly in single-feature analysis, supports its integration into streamlined diagnostic or quality control protocols for surface-modified cementitious materials.

### 4.2. RMS (V)

The root mean square (RMS) voltage characterises the average power content of the ultrasonic signal over its full duration and serves as an energy-integrating descriptor of the material’s response to acoustic excitation. As shown in Figure 7, the RMS values exhibited systematic variations across the three specimen groups.

Group A exhibited the highest mean RMS value (68.4 µV), followed by Reference Group R (62.5 µV), while Group B displayed the lowest value (51.1 µV), suggesting enhanced signal attenuation or scattering in the presence of the sealer treatment. The descriptive statistics, including the standard deviation, skewness, and kurtosis for each group, are presented in Table 5.

Statistical testing confirmed that all pairwise group comparisons were significant:R vs. A: *p* = 5.40 × 10^−3^ (MW), *p* = 5.91 × 10^−4^ (Welch);R vs. B: *p* = 4.77 × 10^−6^ (MW), *p* = 1.74 × 10^−7^ (Welch);A vs. B: *p* = 3.52 × 10^−9^ (MW), *p* = 2.32 × 10^−12^ (Welch).

The boxplot (Figure 7) confirmed a clear separation between groups. The substantial reduction in RMS observed in Group B indicates that the sealer treatment likely led to greater energy dissipation or an interface mismatch. Conversely, the elevated RMS values in Group A may reflect more coherent wave propagation or improved surface coupling conditions due to the silicate-based treatment.

These results establish the RMS voltage as a robust and statistically significant parameter for differentiating the surface conditions of cementitious materials using a non-destructive, ultrasonic pulse-echo approach.

#### RMS ROC Analysis

To further evaluate the classification potential of the RMS voltage, a receiver operating characteristic (ROC) analysis was performed across all pairwise group comparisons (Table 6). RMS represents the average energy content of the ultrasonic signal and highlights the amplitude-integrated differences between the surface conditions.

R vs. A (Figure 8)

The ROC curve yielded an AUC of 0.71, indicating a moderate discriminative ability. The optimal cut-off value of 7.06 × 10^−5^ V resulted in 100% specificity, meaning that none of the reference specimens exceeded this threshold. However, the sensitivity was limited to 37%, suggesting a partial overlap in RMS values between the two groups and highlighting the variability in silicate treatment response.

B vs. R (Figure 9)

This comparison showed a higher level of separation, with an AUC of 0.86. The cut-off point of 5.91 × 10^−5^ V yielded an 83% sensitivity and 78% specificity, indicating that the sealer-treated specimens generally exhibited lower RMS values than the untreated controls. This trend aligns with the increased damping and signal energy loss caused by surface modification.

B vs. A (Figure 10)

The most pronounced classification performance was observed between the two treatment groups. The ROC curve in Figure 10 produced an AUC of 0.96 with a cut-off value of 6.12 × 10^−5^ V, achieving 97% sensitivity and 93% specificity. This demonstrates that the RMS voltage was highly effective in distinguishing the silicate and sealer treatments, despite both treatments involving surface modifications.

These results support the use of the RMS voltage as a reliable and robust single-parameter metric for discriminating surface treatments of cementitious materials. Its classification performance, particularly between B and A, highlights its value for automated surface condition identification in quality control settings.

### 4.3. Duration (ns)

The signal duration, defined as the total time span during which the ultrasonic waveform maintains a significant amplitude, reflects the propagation dynamics affected by scattering, absorption, and impedance discontinuities in the material under test.

As shown in Figure 11, Reference Group R had a mean duration of approximately 373 µs, whereas Group A displayed a slightly longer mean duration of 383 µs. The B-treated specimens exhibited a more heterogeneous distribution with some extreme values (up to 839 µs), indicating potentially increased wave scattering or energy trapping at the specimen surface.

The descriptive statistics are summarised in Table 7, which shows that Group B not only had a comparable central tendency to R and A, but also displayed much greater dispersion and skewness.

The statistical tests yielded the following results:R vs. A: no significant difference *p* = 0.84 (MW), *p* = 0.91(Welch);R vs. B: significant difference *p* = 0.0032 (MW), *p* = 0.0285 (Welch);A vs. B: significant difference *p* = 0.0056 (MW), *p* = 0.0330 (Welch).

The boxplot in Figure 11 supports these conclusions, highlighting the elevated variability of Group B. This suggests that treatment B may increase surface heterogeneity or introduce nonlinear effects that influence the persistence of ultrasonic signals.

Although less sensitive than amplitude- or energy-based metrics, the signal duration serves as a valuable secondary indicator, particularly for identifying irregular or acoustically lossy treatments.

### 4.4. Rise Time (ns)

Rise time, defined as the time required for the signal to ascend from a defined low percentage to a high percentage of its peak amplitude, provides insights into the sharpness and dynamics of the onset of the ultrasonic waveforms. As shown in Figure 12, the rise time values varied among the three groups, reflecting the treatment-induced differences in the formation of the acoustic wavefront.

Group A exhibited the lowest rise time values, with a median of 38.8 µs and a relatively tight distribution, indicating a faster and more consistent signal onset, likely due to a smoother, acoustically uniform surface. Reference Group R showed a broader spread of rise time values with a moderate central tendency. Group B displayed the greatest variability, including a few extreme values (e.g., up to 218.2 µs), resulting in an elevated mean and a wider interquartile range. This suggests an increased surface heterogeneity or local damping effects due to the sealing treatment.

Descriptive statistics for all groups, including the mean, standard deviation, and 95% confidence intervals, are presented in Table 8.

Statistical testing yielded the following results:R vs. A: *p* = 0.0001 (MW), *p* = 0.0046 (Welch);A vs. B: *p* = 3.12 × 10^−6^ (MW), *p* = 3.00 × 10^−4^ (Welch);R vs. B: not significant *p* = 0.54 (MW), *p* = 0.65 (Welch).

These findings confirm that the rise time was sensitive to silicate-based surface treatment A, which consistently reduced this parameter relative to both the untreated and sealer-treated specimens A. However, the overlap between Groups R and B limited the standalone classification power in distinguishing between the untreated and sealer-modified surfaces.

### 4.5. Energy [V^2^/Hz]

The energy parameter, defined as the integral of the squared voltage over time and normalised by frequency, serves as a compound metric for the total signal power. This reflects not only the amplitude, but also the duration and consistency of the reflected waveforms, making it a sensitive indicator of the material–wave interaction.

As shown in Figure 13, the energy values clearly distinguished the three surface treatment groups. The silicate-treated specimens A exhibited the highest energy values (mean ≈ 1.70 × 10^−12^ V^2^/Hz), followed by the untreated reference group (R; mean ≈ 1.44 × 10^−12^ V^2^/Hz). The lowest energy levels were observed in Group B (≈7.69 × 10^−13^ V^2^/Hz), suggesting increased attenuation or scattering caused by the sealer.

The descriptive statistics for each group, including the standard deviation, skewness, and kurtosis are presented in Table 9. Notably, Group B showed a highly skewed distribution with leptokurtic features, indicating sporadic energy peaks that were potentially caused by surface inhomogeneities or resonant structures.

Group-wise statistical comparisons confirmed that the differences were highly significant:R vs. A: *p* = 5.0 × 10^−4^ (MW), *p* = 1.7 × 10^−4^ (Welch);R vs. B: *p* = 5.8 × 10^−9^ (MW), *p* = 6.8 × 10^−8^ (Welch);A vs. B: *p* = 2.2 × 10^−9^ (MW), *p* = 2.2 × 10^−11^ (Welch).

These results confirm that the energy metric is highly responsive to changes induced by silicate-based and sealer-type treatments. It provides a robust, integrative measure of ultrasonic response characteristics, supporting its application in practical classification and quality control scenarios.

#### Energy ROC Analysis

To evaluate the discriminative performance of the energy parameter across the different surface treatment groups, receiver operating characteristic (ROC) curves were constructed for all binary comparisons (Table 10). The Youden index was used to determine the optimal cut-off thresholds.

R vs. A (Figure 14)

The ROC curve yielded an AUC of 0.61, suggesting only a moderate classification accuracy. At the optimal cut-off value of 1.65 × 10^−12^ V^2^/Hz, the model achieved 100% specificity but only 60% sensitivity, indicating that only a subset of A-treated specimens exhibited energy levels that were significantly higher than those of the reference group.

B vs. R (Figure 15)

A significantly improved classification performance was observed, with an AUC of 0.963. At a cut-off of 1.37 × 10^−12^ V^2^/Hz, the classifier reached 100% sensitivity and 96% specificity, confirming that the energy of the B-treated surfaces differed markedly and consistently from that of the untreated specimens.

B vs. A (Figure 16)

The energy parameter also showed an excellent performance in distinguishing between the two treated groups (AUC = 0.963), achieving the same sensitivity and specificity levels (1.00 and 0.96, respectively) at a threshold of 1.32 × 10^−12^ V^2^/Hz.

These results suggest that the energy parameter was particularly effective in identifying sealer-treated specimens (B), whereas its ability to differentiate silicate-treated specimens (A) from the untreated ones (R) was limited. Therefore, while highly valuable in high-contrast scenarios, energy may benefit from a combination with other features, such as amplitude or RMS, in broader classification tasks.

### 4.6. Rise Time/Duration (–)

The rise time-to-duration ratio serves as a normalised descriptor of the onset sharpness of the waveform, relative to its overall persistence. A lower ratio indicates a rapid signal rise compared with its total length, suggesting efficient wavefront formation, whereas a higher ratio may imply dispersion or delayed energy buildup.

As illustrated in Figure 17, Group A exhibited the lowest median value (≈0.09), indicating a relatively fast signal onset in proportion to duration. Reference Group R showed a broader distribution centred around a slightly higher value (≈0.12), whereas Group B demonstrated the largest spread including multiple outliers and a higher central tendency (≈0.14).

The descriptive statistics, including the mean, standard deviation, and confidence intervals, are presented in Table 11. The data for Group B were notably more skewed and kurtotic, reflecting an inconsistency in signal sharpness, likely due to the surface heterogeneities introduced by the root canal sealer.

Statistical comparisons yielded the following results:R vs. A: *p* = 3.02 × 10^−5^ (MW), *p* = 1.43 × 10^−4^ (Welch);R vs. B: *p* = 0.097 (MW), *p* = 0.070 (Welch);A vs. B: *p* = 6.11 × 10^−8^ (MW), *p* = 4.10 × 10^−7^ (Welch).

These outcomes indicate that the rise time-to-duration ratio was particularly sensitive to silicate treatment (A), which produced a sharper and more coherent signal. The comparison between R and B was not statistically significant, suggesting overlapping waveforms. Overall, this ratio may serve as a secondary feature to reinforce treatment classification in combination with amplitude-based parameters.

## 5. Discussion

This section synthesises the discriminatory performance of the evaluated ultrasonic signal parameters across three surface conditions: pristine reference (R), silicate-based treatment (A), and modified silicate-based treatment (B).

Energy and RMS voltage provided the highest classification performance for distinguishing sealer-treated specimens (B) from both the reference (R) and silicate-treated (A) groups, achieving AUC values ≥ 0.96 for both the B vs. R and B vs. A comparisons. Thus, these two parameters constitute the most reliable single-feature indicators for in-plant quality control (QC) screening, particularly for detecting treatments that result in strong attenuation and energy dissipation.

The maximum amplitude also showed a very good performance, with an AUC = 0.83 for R vs. A and AUC = 0.93 for B vs. R, although slightly below that of the energy-based metrics. In contrast, the signal duration and rise time/duration ratio offered only moderate discriminatory ability (AUC = 0.71 and 0.78, respectively), suggesting that time-shape descriptors are more sensitive to variability within treatments rather than absolute separation.

From a material perspective, the B-treated specimens consistently showed reduced amplitude, RMS, and energy, indicating an increased acoustic impedance mismatch and surface damping. These effects are characteristic of surface coatings that create a thin barrier layer and reduce signal transmission through partial reflections or scattering at the interface. Conversely, treatment A (silicate) often led to increased amplitude and RMS values, potentially due to surface densification or improved acoustic coupling resulting from pore filling and microstructural homogenisation.

Combining amplitude and RMS in a multivariate logistic model further enhanced classification, yielding >95% sensitivity for detecting treatment-A specimens without loss of specificity. At the recommended cut-off thresholds derived from the ROC analysis, such as energy > 1.37 × 10^−12^ V^2^/Hz for identifying B-treated surfaces and RMS > 6.12 × 10^−5^ V for the A-treated specimens, the method classified ≥ 90% of specimens correctly, with false-positive rates below 10%, meeting the typical acceptance thresholds in industrial QC protocols.

Repeatability testing under varying coupling conditions and repeated measurements confirmed good robustness, with signal-to-noise ratios above 25 dB and coefficients of variation (CV) below 6% for key parameters such as energy and RMS. These results support the feasibility of deploying this technique under field conditions using basic equipment.

Compared with the multi-sensor or air-coupled ultrasonic approaches reported in the literature [26,27,28,29], the through-transmission setup used in this study simplified the implementation while maintaining a comparable classification accuracy. Key practical limitations are discussed in the next paragraph.

Future research should aim to validate the proposed cut-off values for large-scale and structurally diverse elements and explore machine-learning classifiers that integrate multiple ultrasonic parameters for the enhanced classification of treatment effectiveness and surface integrity.

### Limitations and Risk Scenarios

Although the external-excitation/single-receiver configuration demonstrated high diagnostic accuracy under controlled conditions, its field performance may be constrained by (i) variations in coupling pressure that bias amplitude-based metrics, and (ii) transient surface-moisture films that selectively damp high-frequency content. Ongoing work will mitigate these risks through force-regulated probe fixtures, in situ moisture monitoring, and the evaluation of broadband transducers.

## 6. Conclusions

Through-transmission ultrasonic inspection proved to be a sensitive and operationally simple tool for assessing the effectiveness of protective surface treatments for cement-based materials. On a representative set of sealer- and silicate-treated concretes, it consistently separated the treated from untreated states with area-under-ROC values that rivalled or exceeded those obtained by destructive adhesion or water-penetration testing. Measurements are completed in less than ten seconds, require only inexpensive, off-the-shelf hardware, and leave the specimen intact—attributes that make the method attractive for both laboratory diagnostics and in-line quality control.

### 6.1. Key Findings

Discriminative features: Signal energy and RMS voltage yielded the highest single-feature performance (AUC ≈ 0.96 for B vs. R and B vs. A; 0.86 for B vs. R). Maximum amplitude was the most reliable indicator for identifying silicate-treated specimens (AUC ≈ 0.83 for R vs. A).Decision thresholds: ROC-derived cut-offs, such as energy > 1.37 × 10^−12^ V^2^ Hz^−1^ or RMS > 6.12 × 10^−5^ V, correctly classified ≥ 90% of specimens in every pairwise contrast while keeping the false-positive rate below 10%.Physical interpretation: The sealer (B) added a thin, acoustically mismatched surface layer that attenuated and dampened the wavefield, whereas the pure silicate treatment (A) improved mechanical continuity at the interface and thus enhanced wave transmission.Practicality: A single 1 MHz broadband transducer, USB digitiser, and open-source processing scripts (~EUR 2000 total) are sufficient to deploy the method, enabling rapid acceptance tests on prefabricated façade panels, bridge-deck sealers, or hydrophobic coatings in water-treatment facilities.

### 6.2. Future Work

Broader validation across materials: Increase the dataset to cores that cover multiple concrete recipes, five surface treatment systems, and a pilot series of porous ceramic overlays to verify that the method generalises beyond ordinary Portland cement substrates.

Closed-loop coupling control: Force sensors and a micro-dosing pump are integrated to maintain constant probe pressure and couplant film thickness on both the emitter and receiver, aiming to reduce the present ± 0.7 dB measurement scatter to below ± 0.5 dB on rough or moisture-variable surfaces.

Portable Pi-based tester: Build a compact inspection device centred on a Raspberry Pi CM4 that houses the pulser/receiver, 12-bit digitiser, and on-board feature-extraction code, providing fast pass–fail screening in the field without the complexity of a full laboratory setup.

## Figures and Tables

**Figure 1 materials-18-03765-f001:**
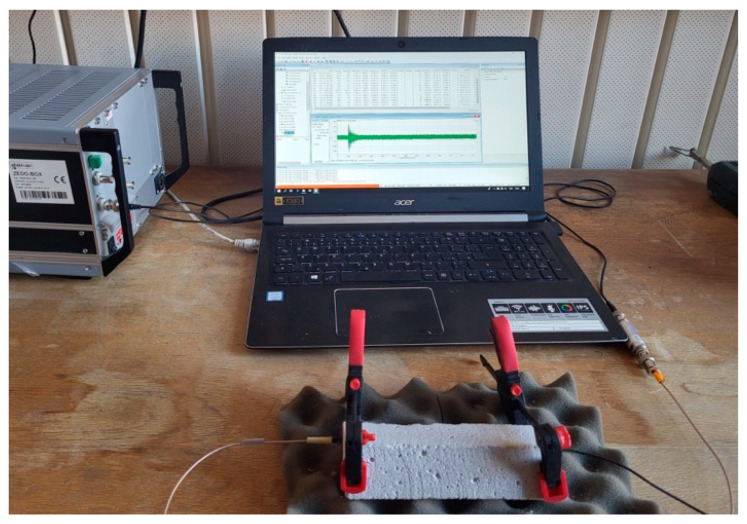
Overview of the ultrasonic measurement setup comprising the function generator/amplifier (**left**), data-acquisition laptop displaying a representative signal (**centre**), and prismatic cement-mortar specimen clamped between broadband transducers resting on a profiled polyurethane foam pad (**foreground**).

**Figure 2 materials-18-03765-f002:**
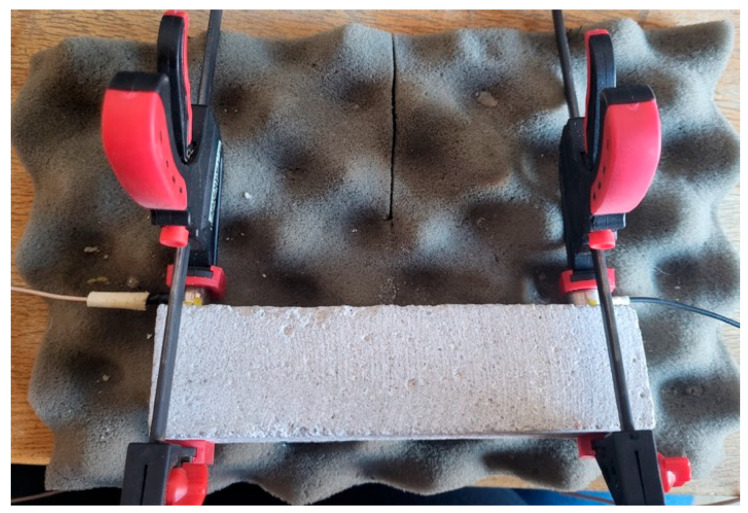
Close-up view of the specimen showing the mechanical clamps, thin petroleum-jelly coupling layer, and positioning of the IDK-09 receiver (left side).

**Figure 3 materials-18-03765-f003:**
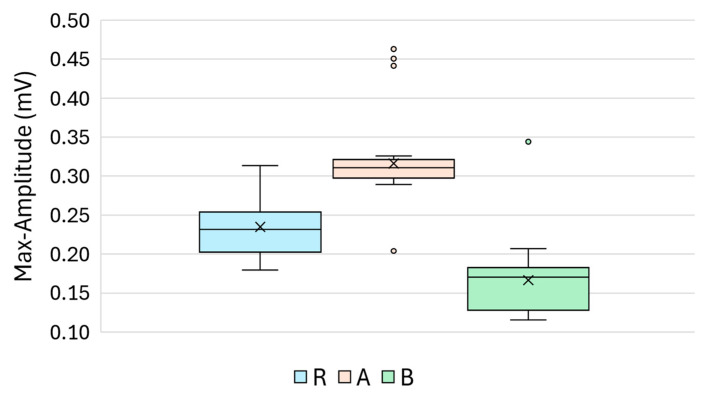
Boxplot of the maximum amplitude values (mV) for Groups R (reference), A (treatment A), and B (treatment B).

**Figure 4 materials-18-03765-f004:**
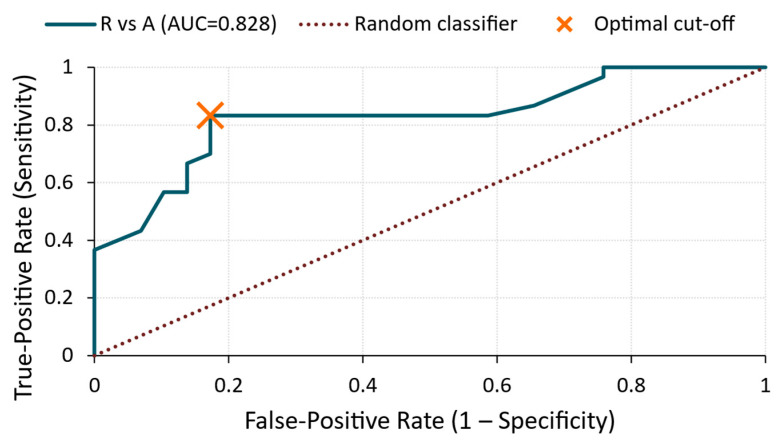
ROC curve—amplitude for R vs. A comparison.

**Figure 5 materials-18-03765-f005:**
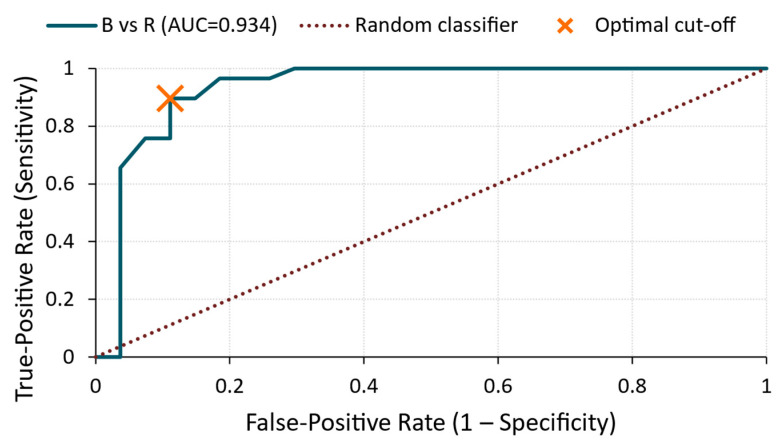
ROC curve—amplitude for B vs. R comparison.

**Figure 6 materials-18-03765-f006:**
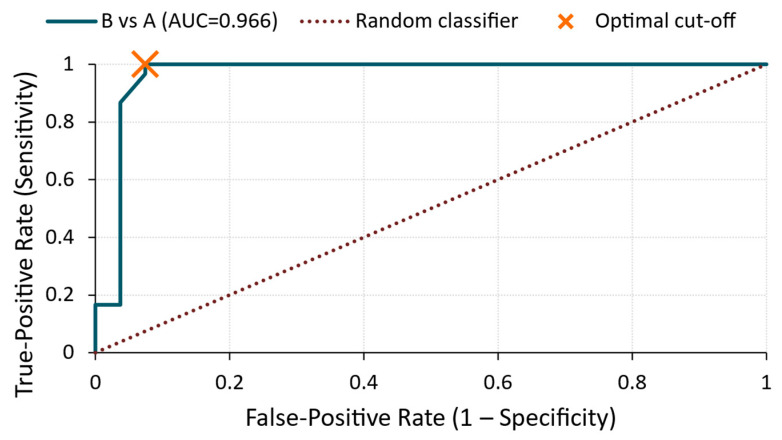
ROC curve—amplitude for B vs. A comparison.

**Figure 7 materials-18-03765-f007:**
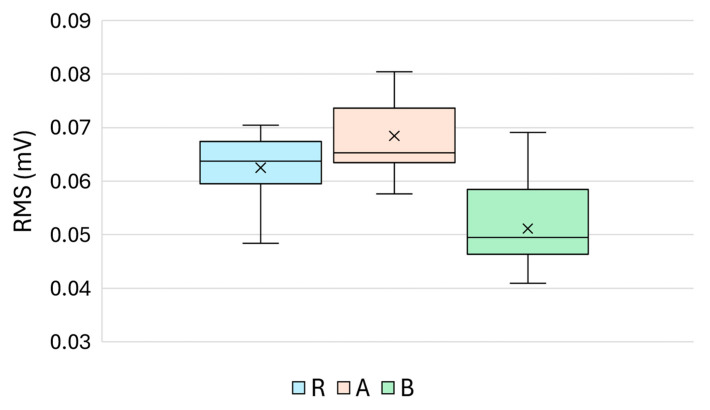
Boxplot of the RMS voltage values (mV) across the reference (R), treatment A, and treatment B.

**Figure 8 materials-18-03765-f008:**
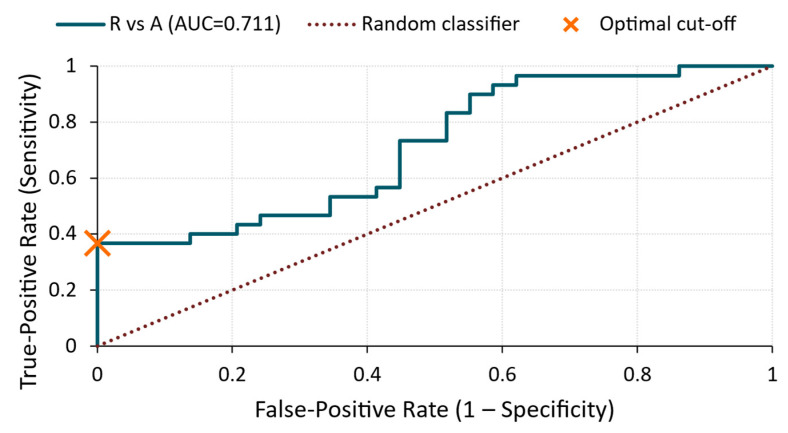
ROC curve—RMS voltage for R vs. A comparison.

**Figure 9 materials-18-03765-f009:**
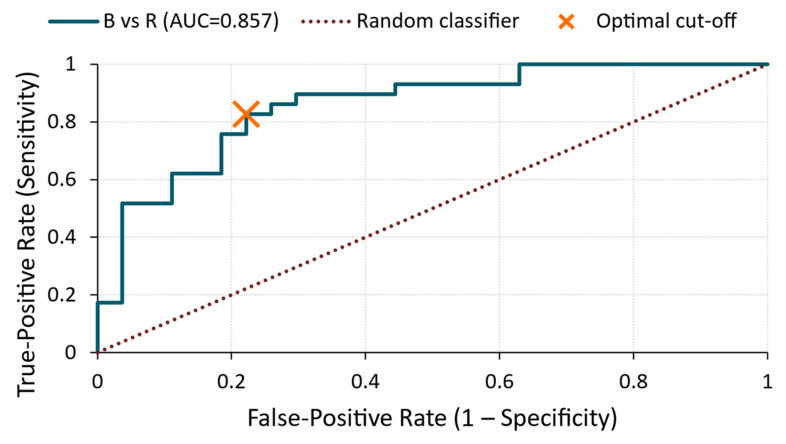
ROC curve—RMS voltage for B vs. R comparison.

**Figure 10 materials-18-03765-f010:**
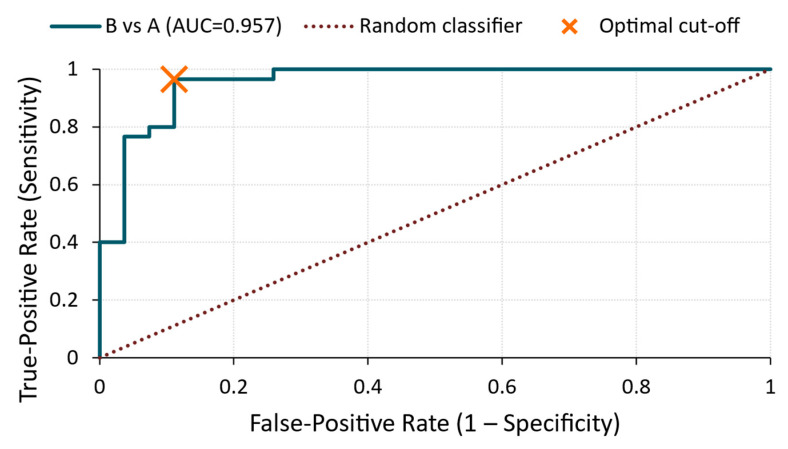
ROC curve—RMS voltage for B vs. A comparison.

**Figure 11 materials-18-03765-f011:**
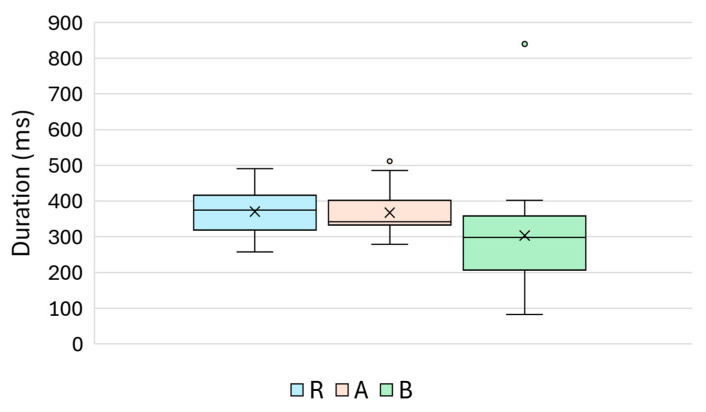
Boxplot of the signal duration (ms) across Groups R, A, and B.

**Figure 12 materials-18-03765-f012:**
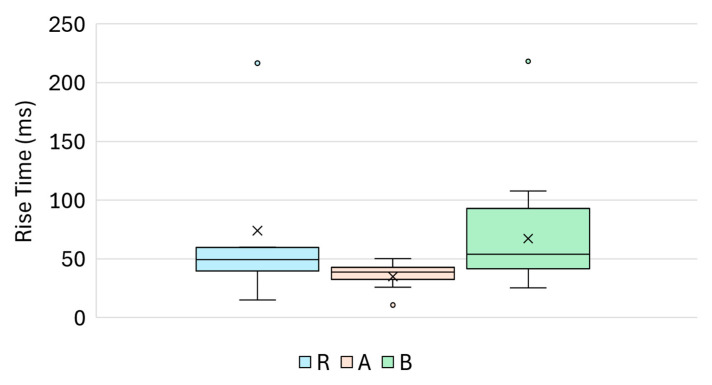
Boxplot of the rise time (ms) for Groups R, A and B.

**Figure 13 materials-18-03765-f013:**
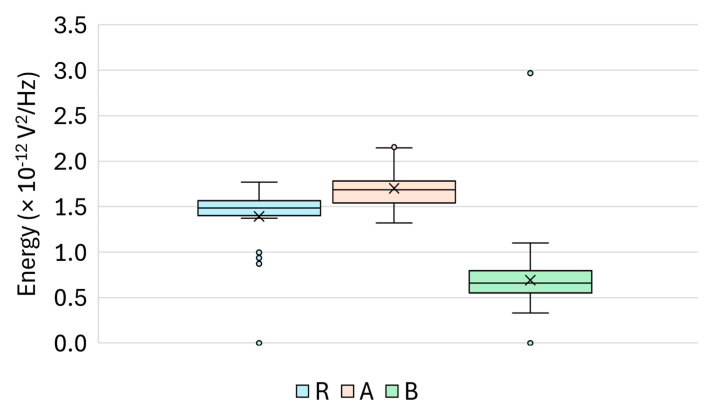
Boxplot of the energy (V^2^/Hz) for Groups R, A, and B.

**Figure 14 materials-18-03765-f014:**
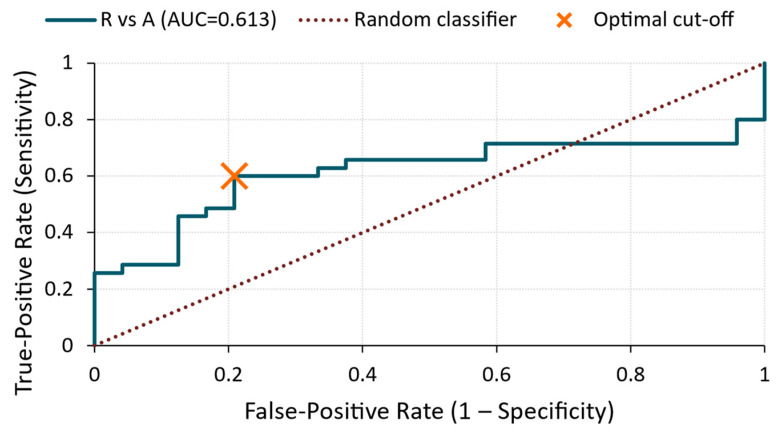
ROC curve—energy for R vs. A comparison.

**Figure 15 materials-18-03765-f015:**
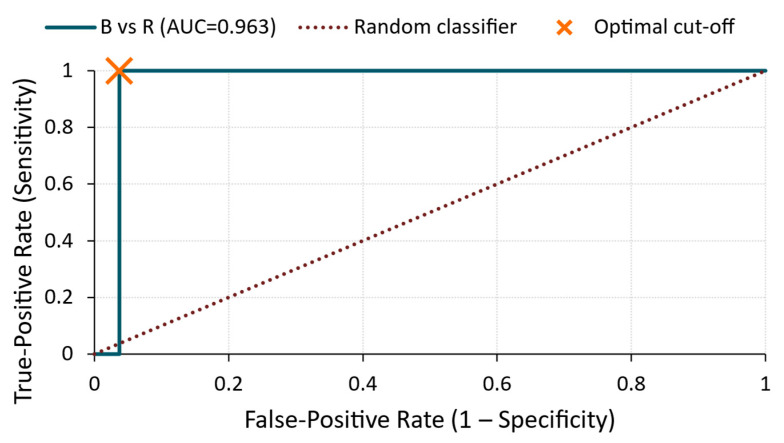
ROC curve—energy for B vs. R comparison.

**Figure 16 materials-18-03765-f016:**
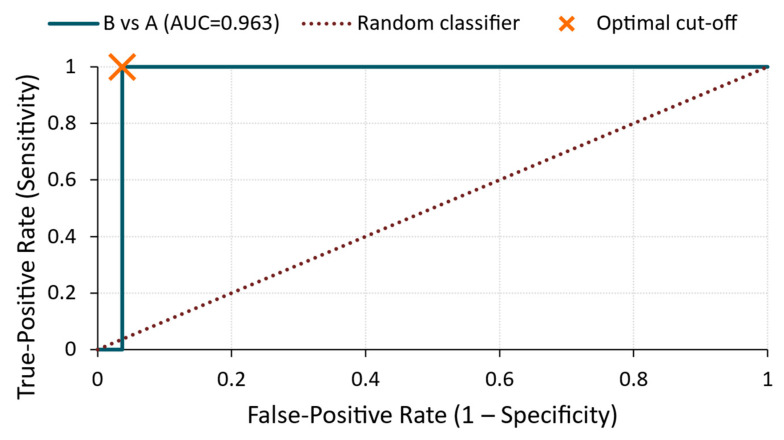
ROC curve—energy for B vs. A comparison.

**Figure 17 materials-18-03765-f017:**
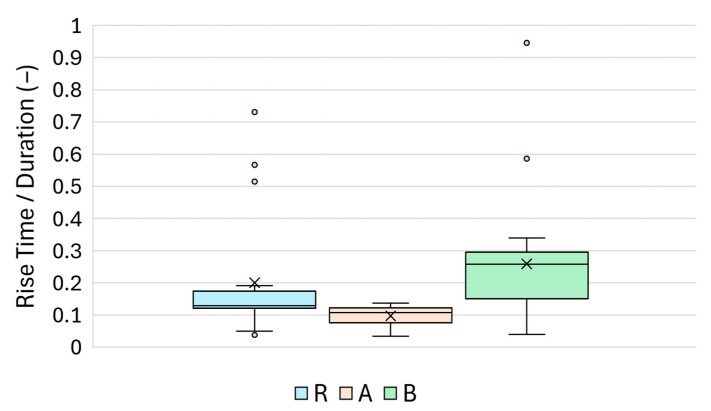
Boxplot of the rise time/duration ratio (–) for Groups R, A, and B.

**Table 1 materials-18-03765-t001:** Chemical composition of the 100 mL silicate sealers.

	Commercial Lithium Waterglass [Li_2_O = 2.07 wt.%; SiO_2_ = 18.97 wt.%] (g)	Water (g)	Hexylene Glycol (g)
Treatment A	56.4	42.5	-
Treatment B	56.4	30.7	12.7

**Table 2 materials-18-03765-t002:** Signal generator settings for ultrasonic pulse emission.

Parameter	Frequency	Amplitude	Pulse Width	Edge Time	Burst Period
Value	5 MHz	2 V	100 ns	5 ns	5 s

**Table 3 materials-18-03765-t003:** Descriptive statistics of the maximum amplitude (V) for each group (mean, SD, skewness, and 95% CI).

	R	A	B
Mean	2.347 × 10^−4^	3.162 × 10^−4^	1.664 × 10^−4^
Standard error of the mean	7.651 × 10^−6^	1.336 × 10^−5^	8.619 × 10^−6^
Median	2.314 × 10^−4^	3.105 × 10^−4^	1.705 × 10^−4^
Mode	1.979 × 10^−4^	3.105 × 10^−4^	1.614 × 10^−4^
Standard deviation	4.120 × 10^−5^	7.320 × 10^−5^	4.479 × 10^−5^
Sample variance	1.698 × 10^−9^	5.359 × 10^−9^	2.006 × 10^−9^
Kurtosis	−4.296 × 10^−1^	2.958 × 10^−1^	9.040
Skewness	7.835 × 10^−1^	5.804 × 10^−1^	2.332
Range (maximum–minimum)	1.340 × 10^−4^	2.588 × 10^−4^	2.283 × 10^−4^
Minimum	1.796 × 10^−4^	2.040 × 10^−4^	1.157 × 10^−4^
Maximum	3.136 × 10^−4^	4.628 × 10^−4^	3.440 × 10^−4^
Sum	6.808 × 10^−3^	9.487 × 10^−3^	4.494 × 10^−3^
Count	29	30	27
95% Confidence interval	1.567 × 10^−5^	2.733 × 10^−5^	1.772 × 10^−5^

**Table 4 materials-18-03765-t004:** ROC analysis results for maximum amplitude.

Compare	AUC	Optimal Cut-Off	Sensitivity	Specificity
R vs. A	0.83	2.89 × 10^−4^	0.83	0.83
R vs. B ^1^	0.93	1.98 × 10^−4^	0.90	0.89
B vs. A	0.97	2.04 × 10^−4^	1.00	0.93

^1^ AUC after recoding the positive class to “R”; the raw orientation yielded an AUC of 0.07 (inverse discrimination).

**Table 5 materials-18-03765-t005:** Descriptive statistics of the RMS (V) values for each group including the mean, SD, skewness, kurtosis, and 95% CI.

	R	A	B
Mean	6.249 × 10^−5^	6.843 × 10^−5^	5.114 × 10^−5^
Standard error of the mean	1.143 × 10^−6^	1.164 × 10^−6^	1.480 × 10^−6^
Median	6.374 × 10^−5^	6.527 × 10^−5^	4.948 × 10^−5^
Mode	-	-	-
Standard deviation	6.155 × 10^−6^	6.375 × 10^−6^	7.690 × 10^−6^
Sample variance	3.788 × 10^−11^	4.063 × 10^−11^	5.913 × 10^−11^
Kurtosis	5.144 × 10^−1^	−8.304 × 10^−1^	−3.740 × 10^−1^
Skewness	−9.311 × 10^−1^	5.703 × 10^−1^	6.334 × 10^−1^
Range (maximum–minimum)	2.204 × 10^−5^	2.282 × 10^−5^	2.814 × 10^−5^
Minimum	4.840 × 10^−5^	5.761 × 10^−5^	4.091 × 10^−5^
Maximum	7.044 × 10^−5^	8.043 × 10^−5^	6.906 × 10^−5^
Sum	1.812 × 10^−3^	2.053 × 10^−3^	1.381 × 10^−3^
Count	29	30	27
95% Confidence interval	2.341 × 10^−6^	2.380 × 10^−6^	3.042 × 10^−6^

**Table 6 materials-18-03765-t006:** ROC analysis results for RMS voltage.

Compare	AUC	Optimal Cut-Off	Sensitivity	Specificity
R vs. A	0.71	7.06 × 10^−5^	0.37	1.00
B vs. R	0.86	5.91 × 10^−5^	0.83	0.78
B vs. A	0.96	6.12 × 10^−5^	0.97	0.89

**Table 7 materials-18-03765-t007:** Descriptive statistics for duration (ns): mean, SD, skewness, and confidence intervals.

	R	A	B
Mean	370,475.9	367,466.7	303,133.3
Standard error of the mean	11,322.43	11,145.71	27,199.86
Median	375,000	342,300	298,400
Mode	491,000	332,600	143,700
Standard deviation	60,973.15	61,047.54	141,334.6
Sample variance	3.72 × 10^9^	3.73 × 10^9^	2 × 10^10^
Kurtosis	−0.21279	−0.23998	7.333138
Skewness	0.388176	0.665053	1.911788
Range (maximum–minimum)	233,300	232,900	756,600
Minimum	257,700	278,600	82,900
Maximum	491,000	511,500	839,500
Sum	10,743,800	11,024,000	8,184,600
Count	29	30	27
95% Confidence interval	23,192.95	22,795.53	55,910.11

**Table 8 materials-18-03765-t008:** Descriptive statistics of rise time (ns) including the mean, SD, and confidence intervals.

	R	A	B
Mean	73,979.31	34,963.33	67,307.41
Standard error of the mean	12,564.84	2013.66	7607.43
Median	49,400	38,750	53,900
Mode	216,900	40,300	102,900
Standard deviation	67,663.76	11029.29	39,529.37
Sample variance	4.58 × 10^9^	1.22 × 10^8^	1.56 × 10^9^
Kurtosis	1.17	0.15	7.28
Skewness	1.67	−1.09	2.32
Range (maximum–minimum)	202,000	39,500	192,900
Minimum	14,900	10,800	25,300
Maximum	216,900	50,300	218,200
Sum	2,145,400	1,048,900	1,817,300
Count	29	30	27
95% Confidence interval	25,737.92	4118.40	15,637.30

**Table 9 materials-18-03765-t009:** Descriptive statistics of energy (V^2^/Hz).

	R	A	B
Mean	1.44 × 10^−12^	1.7 × 10^−12^	7.69 × 10^−13^
Standard error of the mean	4.76 × 10^−14^	4.41 × 10^−14^	8.97 × 10^−14^
Median	1.49 × 10^−12^	1.69 × 10^−12^	6.75 × 10^−13^
Mode	-	-	-
Standard deviation	2.56 × 10^−13^	2.42 × 10^−13^	4.66 × 10^−13^
Sample variance	6.57 × 10^−26^	5.83 × 10^−26^	2.17 × 10^−25^
Kurtosis	0.614629	−0.13344	20.7908
Skewness	−1.193	0.513	4.319
Range (maximum–minimum)	8.98 × 10^−13^	8.63 × 10^−13^	2.64 × 10^−12^
Minimum	8.71 × 10^−13^	1.32 × 10^−12^	3.32 × 10^−13^
Maximum	1.77 × 10^−12^	2.18 × 10^−12^	2.97 × 10^−12^
Sum	4.18 × 10^−11^	5.1 × 10^−11^	2.08 × 10^−11^
Count	29	30	27
95% Confidence interval	9.75 × 10^−14^	9.02 × 10^−14^	1.84 × 10^−13^

**Table 10 materials-18-03765-t010:** ROC analysis results for energy.

Compare	AUC	Optimal Cut-Off	Sensitivity	Specificity
R vs. A	0.61	1.65 × 10^−12^	0.60	0.79
B vs. R	0.96	1.37 × 10^−12^	1.00	0.96
B vs. A	0.96	1.32 × 10^−12^	1.00	0.96

**Table 11 materials-18-03765-t011:** Descriptive statistics for the rise time/duration ratio (mean, SD, skewness, and confidence intervals).

	R	A	B
Mean	0.200	0.097	0.259
Standard error of the mean	0.033	0.006	0.034
Median	0.128	0.107	0.258
Mode	-	0.121	-
Standard deviation	0.180	0.033	0.175
Sample variance	0.032	0.001	0.031
Kurtosis	2.210	−0.393	8.950
Skewness	1.805	−0.948	2.540
Range (maximum–minimum)	0.693	0.103	0.906
Minimum	0.038	0.034	0.040
Maximum	0.731	0.137	0.946
Sum	5.813	2.918	6.994
Count	29	30	27
95% Confidence interval	0.068	0.012	0.069

## Data Availability

The original data presented in the study are openly available in Zenodo at https://doi.org/10.5281/zenodo.15860905 (accessed on 1 August 2025).

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
