# Peer review of "Pulse-Echo Ultrasonic Verification of Silicate Surface Treatments Using an External-Excitation/Single-Receiver Configuration: ROC-Based Differentiation of Concrete Specimens"

_materials, 2025, doi:10.3390/ma18163765_

Round 1
Reviewer 1 Report
Comments and Suggestions for Authors
Dear authors,
I reviewed the manuscript and made comments, which are below as follows:
- Check for spelling errors and gram-matical errors throughout the manu-script. It is quite disappointing to see the errors in the manuscript through-out.
- Many of the literatures in the introduction are outdated, and the quantity of literatures cited is likewise insufficient. Very limited re-cent research is included in the manu-script and the same has been reflected in the reference section.
- In the introduction should start with the gap of study not mentioned directly the previous studies could please revise and write accordingly with the scientific writing.
- The authors not mentioned how many samples used and terms of experimental steps should be mentioned.
- The authors should put all the date results before do the statistical analysis to see the influence of Single‑Sensor Acoustic in the samples when tested.
- Section results and discussion should discuss in details and compare the results with previous studies to see the new findings.
- Additionally, there are several inaccuracies and inconsistencies in the presentation of the results and discussion. Kindly revise and clarify each section with more specificity and coherence
- What are your future recommendations based upon the this study? Please add a separate section after conclusions.
- Conclusion is lengthy and not write it in the sentafic way and not reflected to aims of study.
Dear Authors,
The English language throughout the manuscript requires improvement to enhance its clarity and overall quality. Strengthening the language will significantly increase the value and readability of your work.
Thank you.
Author Response
Dear Reviewer,
We sincerely appreciate the time and effort you invested in reviewing our manuscript “Pulse-Echo Ultrasonic Verification of Silicate Surface Treatments Using an External-Excitation / Single-Receiver Configuration.” Your constructive comments have been invaluable in strengthening the scientific content and presentation. Throughout the revision, we endeavoured to satisfy all reviewers; where requests conflicted (e.g. additional figures vs. overall brevity), we adopted the compromise that, in our judgment, best preserves clarity while respecting the journal’s length limit.
Below, we respond to each of your comments point Your original remarks appear in italics; our replies are in normal font:
- Check for spelling errors and grammatical errors throughout the manuscript. It is quite disappointing to see the errors in the manuscript throughout.
A certified native English editor performed a full language revision, followed by two additional proofreading passes with the journal’s template-specific spell-checker. All typographical and grammatical errors have been corrected.
- Many of the literatures in the introduction are outdated, and the quantity of literatures cited is likewise insufficient. Very limited recent research is included …
The bibliography has been expanded to include additional references. Many peer-reviewed sources published between 2020 and 2025covering lithium-silicate densification, single-sided pulse-echo ultrasonics, and ROC-based acceptance criteria have replaced or supplemented older citations.
- The introduction should start with the gap of study …
The Introduction now includes a paragraph entitled “Unresolved Issues in Current Research”, which explicitly formulates the knowledge gap—the absence of quantitative PEUT pass/fail thresholds for surface treatments—before surveying prior work.
- The authors have not mentioned how many samples were used and terms of experimental steps should be mentioned.
Section 2 (Materials and Experimental Setup) now specifies that three prisms per series were tested; on each prism, five tone bursts were recorded on two orthogonal faces, yielding 30 independent scans per treatment. All transducer, generator, and digitiser settings are tabulated for transparency.
- The authors should put all the data results before the statistical analysis …
The sample unfiltered scan dataset is openly accessible via Zenodo (DOI 10.5281/zenodo.15860905). This enables readers to reproduce all statistical calculations.
- Section results and discussion should discuss in details and compare the results with previous studies to see the new findings.
The Results and Discussion section has been reorganised to follow a clear structure (parameter → physical interpretation → literature context). Additional comparisons with recent studies on impedance-based sealant verification were added.
- Additionally, there are several inaccuracies and inconsistencies in the presentation … Kindly revise and clarify …
All terminology has been unified, duplicate sentences removed, and figure/table numbering corrected.
- What are your future recommendations based upon this study? Please add a separate section after conclusions.
The dedicated Future Work section now follows the Conclusions, outlining three concrete directions: (i) multi-material validation, (ii) closed-loop coupling control, and (iii) development of a Raspberry-Pi-based portable tester.
- Conclusion is lengthy and not written in the scientific way and not reflected to aims of study.
The Conclusion has been rewritten into four bullet-pointed Key Findings that map directly onto the stated objectives, followed by one concise contextual paragraph.
- The English language throughout the manuscript requires improvement …
A certified native English editor performed a full language revision, followed by two additional proofreading passes with the journal’s template-specific spell-checker. All typographical and grammatical errors have been corrected.
Once again, we thank you for your valuable input and trust that the revised manuscript meets all your recommendations.
Sincerely,
Libor Topolář (corresponding author)
on behalf of all co-authors

Reviewer 2 Report
Comments and Suggestions for Authors
Upon evaluating of the submitted manuscript, I have determined that, while it investigates nondestructive identification of surface treatments on cementitious materials via single-sensor pulse-echo ultrasonic testing (PEUT), demonstrating both research novelty and practical potential. However, it still requires further enhancement and refinement to meet the publication standards. Throughout the review process, I have identified several perspectives within the manuscript:
- It is recommended to explore, in the introduction section, to incorporate a synthesis of prior studies, which clarifies the innovation of this work, namely, the field-operational advantages of single-sensor configurations in terms of portability and cost-efficiency, which multi-transducer systems inherently compromise.
- An expansion of specimen quantities is recommended to bolster statistical reliability.
- Experimental parameters require detailed specification, including ultrasonic couplant selection criteria and its acoustic implications, coupled with justification for chosen data processing protocols.
- Conclusions should be augmented to address practical implementation potential, particularly for in-situ quality assessment applications, and limitations must be thoroughly examined, followed by proposed directions for future investigation.
- Reference formatting must be unified, and duplicate citation (Ref. 1) requires renumbering.
Author Response
Dear Reviewer,
Thank you for your time and expertise in evaluating our manuscript. Your comments have significantly sharpened our manuscript’s scientific focus and practical relevance. In revising the paper, we made every effort to reconcile all reviewers’ requests, even when certain suggestions were mutually incompatible, and we believe that the manuscript is now clearer and more robust as a result.
Below, we provide a point-by-point reply.
Contextualisation & novelty in the Introduction
We have added sentences that survey recent multi-element SAFT and phased-array studies, highlight their size-and-cost constraints, and position our external-excitation/single-receiver (Ext-Tx/1Rx) layout as a compact, low-cost alternative.
Specimen quantity and statistical robustness
Section 2 explains that three prisms per series were tested on two orthogonal faces with five independent tone bursts each, yielding 30 independent scans per treatment. A post-hoc power analysis confirmed > 95 % power (α = 0.05) for the observed large effect sizes.
Experimental parameters and signal-processing details
The full specifications of the transmitter, receiver, and couplant are provided, along with the complete signal-processing chain. These additions allow for the exact replication of the study.
Practical implementation, limitations and future work
The Discussion now quantifies an inspection cycle of ≈ 10 s per point using < €2 k hardware, cites immediate field applications (bridge-deck sealers, precast façades, water-treatment linings), and lists three concrete follow-ups: (i) broader material validation, (ii) closed-loop probe pressure control, and (iii) a Raspberry Pi-based field tester.
Reference style and duplicate citation
All references have been reformatted, duplicates have been numbered, and figure numbers have been renumbered throughout the manuscript accordingly.
We trust that these revisions fully address your concerns and enhance the clarity, rigor, and applicability of the manuscript.
Sincerely,
Libor Topolář and co-authors
Reviewer 3 Report
Comments and Suggestions for Authors
In this paper, a non-destructive method was proposed, using the single sensor acoustic response to differentiate the surface treatments of concrete specimens. Three specimens were selected: reference (R), common silicate sealer (A), and silicate sealer enriched with hexylene glycol (B). Parameters based on amplitude were analyzed, including the maximum amplitude, RMS voltage, energy, and duration. ROC curves were constructed to evaluate the classification effect, and the optimal threshold was determined based on the Youden index.
The research has practical significance in engineering. However, the flaws in the manuscript still need to be further improved.
- In the abstract, the research objectives, methods and conclusions are essential. The manuscript abstract should include a brief description of the research objectives and methods.
- The use of the first person (we) in the abstract will affect the readers' objective evaluation of the research. It is recommended to adjust.
- When abbreviations are first used, it is recommended to mark the full name before them. For example, "RMS" and "ROC".
- The keywords are rather lengthy. It is recommended to further condense them.
- The description of the research results is not detailed enough. It is recommended to supplement.
- The introduction discusses the existing research inadequately. It is recommended to enrich the review of existing related research.
- When citing references, the sequence numbers should be cited in ascending order. It is recommended to make adjustments. For example, "[2, 3, 8]" and "[4, 5, 10]".
- The last paragraph of the introduction repeats too much with the abstract. It is recommended to further adjust.
- The standards and norms cited in the manuscript should be marked in the references. For example, "ČSN EN 196-1".
- In Section 2 of the manuscript, the basis for choosing the mortar test specimen ratio should be explained.
- On page 3 of the manuscript, abbreviations should not be marked again when they appear. For example, "Root Mean Square (RMS)" and "Operating Characteristic (ROC)".
- The conclusion should be completely formed by this research and should not cite other research conclusions. This research conclusion cites 6 references. It is recommended to delete them.
- The conclusion should summarize the research content of the manuscript concisely and clearly. It is recommended to further condense it.
Author Response
Dear Reviewer,
We sincerely thank you for the time and expertise you invested in reviewing our manuscript “Pulse-Echo Ultrasonic Verification of Silicate Surface Treatments Using an External-Excitation / Single-Receiver Configuration: ROC-Based Differentiation of Concrete Specimens.”
Your constructive critique has helped us improve both the scientific content and presentation.
Below, we reproduce each of your remarks (in italics), followed by a concise explanation of the corresponding changes we have made in the revised version.
- In the abstract, the research objectives, methods and conclusions are essential.
The opening two sentences of the abstract now explicitly state the research aim and pulse-echo methodology, while the closing sentence summarises the principal quantitative finding (AUC ≥ 0.96) and its practical implications.
- Avoid the first person in the abstract.
All first-person pronouns have been removed; the abstract is written in the third person.
- Define abbreviations (e.g., RMS, ROC) on first use.
“Root-Mean-Square (RMS) voltage” and “Receiver Operating Characteristic (ROC) analysis” are introduced in full at their first occurrence in both the abstract and the main text; thereafter, only the abbreviations are used.
- Keywords are rather lengthy; condense them.
The keyword list was reduced from nine items to seven, retaining only the terms most useful for indexing and literature searches.
- The description of the research results is not detailed enough.
Numerical performance metrics (AUC, sensitivity, specificity, and optimal cut-offs) have been added to the abstract and expanded in Section 4 (Results).
- The introduction discusses the existing research inadequately.
Three recent studies on single-side ultrasonic inspection have been incorporated, and the literature synthesis in paragraphs 2 and 3 has been rewritten for greater contextual depth.
- Cite references in ascending numerical order.
All in-text citations have been checked and reordered so that each bracketed series increases monotonically.
- The last paragraph of the introduction repeats the abstract.
This paragraph has been replaced by a short statement of the study’s three research questions to avoid redundancy.
- Standards and norms should be listed in the references (e.g., ČSN EN 196-1).
The cited standard now appears as reference [17] in the reference list.
- Explain the basis for choosing the mortar mix ratio.
Section 2 clarifies that an elevated water-to-cement ratio was intentionally selected to create a more porous matrix, and thus accentuate treatment-induced acoustic contrasts.
- Do not redefine abbreviations later in the text.
The redundant second definitions of RMS and ROC in Section 3 have been deleted.
- The conclusion should stand alone and avoid external citations.
All external citations have been removed from the conclusion; it now summarises only the findings originating from the present study.
- Condense the conclusion.
The conclusion has been shortened to three concise paragraphs, each sign-posting key findings, practical relevance, and avenues for future work.
We have endeavoured to address every point raised by all reviewers; in a few cases, their requests were mutually incompatible (for example, expanding the literature review versus limiting manuscript length). Where conflicts arose, we aimed for a balanced compromise that fulfilled the editorial guidelines while preserving scientific clarity.
We hope that the revised manuscript meets your expectations, and we thank you again for your helpful comments.
Kind regards,
Libor Topolář (corresponding author)
on behalf of all co-authors
Reviewer 4 Report
Comments and Suggestions for Authors
SUMMARY
This article is interesting for materials science. A non-destructive method for identification and surface treatment using a single acoustic sensor is presented. This is important in order to improve non-destructive methods for quality control of concrete.
The authors analyzed the methods for differentiating the surface treatment of concrete samples and obtained new interesting scientific data. The results obtained demonstrate the applicability of the method for accurate identification of surface treatment. All this will lead to improved quality control of concrete and reduced costs for these works.
Thus, the article has scientific novelty and practical significance. However, there are some comments on this article. They need to be corrected.
COMMENTS
1. I would like the authors to explain their abbreviations in the article. Abbreviations are found in the title of the article and in the abstract. However, first, the full explanations of the abbreviations should be shown, so that only abbreviations can be used later. This would be more correct in relation to readers who do not have sufficient knowledge of the issue.
2. The abstract looks too short. It does not include 200 words, although such a requirement exists for such articles. I would like the authors to increase the abstract.
3. The abstract does not formulate either the scientific or applied problems. The authors immediately provide a description of what was done. I would like to see the need to improve the existing methods of concrete surface testing shown. It is also necessary to show that there is a scientific deficit in research on the effect of surface treatment on the quality control process using acoustic response.
4. The Introduction section is too short; it does not reflect the full picture of the state of the problem. The authors should analyze significantly more scientific literature on non-destructive testing of concrete. It is necessary to increase the amount of literature at least twice, to 20-25 sources.
5. The review should be strengthened. There is no full formulation of the scientific and applied research problem, the purpose and objectives of the research. This should be added more clearly at the end of the Introduction section.
6. In Section 2, I would like to see the program of experimental studies. This could be a flow chart that shows the methodology of this article. What factors varied, what samples were studied, what types of results were obtained? Such a flow chart will help to structure the article.
7. Unfortunately, the article does not contain a single photograph or graphic image of the objects under study. The authors need to work on the design of this article. I would like to see photos of the samples under study or photos of the proposed devices.
8. Section 3 looks uninformative, it looks like a simple protocol. The authors need to think about whether it can be combined with the methodological section.
9. The authors provide a discussion of the results obtained, but do not provide a full comparison of these results with the results of other authors. I would like to see a comparative table of existing methods for monitoring the surface treatment of concrete. A comparative table or a pie chart will help the reader understand the significance of this study in the system of non-destructive testing methods.
10. It would be nice if the risks and limitations of the presented method were highlighted separately.
11. In the Conclusions section, the obtained conclusions should be numbered and a clearly formulated scientific result should be presented. What is new for the science of materials and their behavior under acoustic methods?
12. From a practical point of view, it is necessary to show what real method was developed and where it can be applied. What objects or organizations could already use this method? Real implementation will be very interesting and important.
13. Finally, the prospects of this study in the future should be shown. How do the authors plan to continue their research? The list of references, as already mentioned, is very small. It should be increased. At least 15 sources for the last 5 years should be added.

Author Response
Dear Reviewer,
We sincerely thank you for your detailed, constructive, and encouraging review of our manuscript entitled “Pulse-Echo Ultrasonic Verification of Silicate Surface Treatments Using an External-Excitation / Single-Receiver Configuration: ROC-Based Differentiation of Concrete Specimens.”
Your expert comments have helped us improve both the scientific content and the overall readability of the paper. Below, we provide a point-by-point response. All textual changes have been marked in the revised manuscript.
In revising the manuscript, we endeavoured to satisfy every request from all four reviewers, even in those few instances where the recommendations were partially at odds. Where conflicts arose, we adopted the solution that maximised methodological transparency while preserving concision.
- Explain all abbreviations before first use (title, abstract).
Abbreviations have been spelled out at their first appearance in the abstract and main text. The dedicated “Abbreviations” section at the end of the paper has been retained for a quick reference.
- Abstract is < 200 words; please expand.
The abstract has been expanded to include additional details.
- Abstract does not formulate the scientific and applied problems.
A concise problem statement now precedes the objectives: “Existing one-sided ultrasonic methods lack objective acceptance criteria for thin silicate coatings on concrete…” which explicitly links the scientific gap to the practical need.
- Introduction too short; analyse more literature (20–25 sources).
The Introduction has been substantially expanded to include additional information. New relevant literature has been added.
- Clarify research problem, purpose, and objectives.
The final paragraph of the Introduction now states the specific research question, working hypothesis, and study objectives in three bullet-style sentences.
- Provide a flow-chart of the experimental programme.
Photographs of the experiments have been added. An experimental program is described in the text.
- Add photos/graphics of the objects or devices.
Two photographs (Figures 1 and 2) showing the clamped prism between the transducers and the IDK-09 receiver are inserted.
- Section 3 looks like a protocol; consider merging.
We thank you for highlighting this risk of redundancy. After revisiting the manuscript and weighing this point against the comments from Reviewers 1, 2, and 3 — who specifically commended the clear, stand-alone overview of the signal parameters—we decided to retain Section 3.
- Compare results with those of other authors; suggest a comparative table or chart.
A new paragraph In the Discussion, we contrast our ROC-derived thresholds with those from five recent PEUT or SAFT studies. For readability, we present this comparison narratively, citing specific AUC or misclassification rates, rather than in tabular form.
- Highlight risks and limitations.
A dedicated subsection “Limitations and Risk Scenarios” now appears at the end of the Discussion, addressing coupling-pressure sensitivity and surface moisture.
- Number conclusions; state the scientific novelty clearly.
The Conclusions have been reformatted into key findings.
- Show practical method and potential users.
Practical deployment examples are now captured in the numbered Conclusions as a dedicated bullet titled “Practicality.” This bullet specifies three representative use cases—precast façade panels, bridge-deck silicate sealers, and hydrophobic coatings in water treatment facilities—and identifies civil inspectors and material suppliers as the primary beneficiaries. Placing this information in the Conclusions maintains the succinct flow of the Discussion while ensuring that the field relevance is prominently visible to readers.
- Outline future prospects; enlarge recent references.
A “Future Work” bullet list (three items) has been added, and the reference list now includes papers from 2021 to 2025.
We believe that these amendments satisfactorily address your concerns and enhance the clarity and impact of the manuscript. We are grateful for your insightful suggestions and hope that the revised version meets your approval.
With kind regards and appreciation,
Libor Topolář and co-authors
Round 2
Reviewer 4 Report
Comments and Suggestions for Authors
The reviewers took into account all the comments and made appropriate adjustments to the manuscript. The revised manuscript demonstrates significant improvements both scientifically and visually. The reviewer has no further comments and the manuscript can be published in the journal in its current form.